# Associations between self-rated health and depressive symptoms among middle-aged and older adults in China: A cross-lagged panel analysis (2011–2020)

**Shuyan Leng, Lihua Yao**[ID]**\*and Jiajia Deng**

College Of Public Health, Chongqing Medical University, Chongqing, China

\* 100179@cqmu.edu.cn

## Abstract

### Background

As the aging of the population accelerates, mental health is important to the quality of life of middle-aged and older people, as well as to the healthcare system and the economy. This study focuses on the relationship between depressive symptoms and self-rated health in middle-aged and older people, as well as whether this relationship differs between urban and rural China.

### Methods

Data from the China Health and Retirement Longitudinal Study (CHARLS). The final analysis included 10503 participants aged 45 years or above. The longitudinal cross-lagged panel analysis was used to assess the relationship between self-rated health and depressive symptoms.

### Results

The model revealed a bidirectional relationship between self-rated health and depressive symptoms. Self-rated health at each time point had a significant effect on subsequent depressive symptoms, and vice versa. However, no urban-rural disparities in these relationships.

### Conclusion

Self-rated health and depressive symptoms were interrelated over time among middle-aged and older adults. To ensure the mental health and quality of life in later life of middle-aged and older adults, it is essential to assess and manage both self-rated health and depressive symptoms.

**Data availability statement:** All data collected by the project are maintained at the Peking University Institute for Social Sciences Research and published on the CHARLS project (http://charls.pku.edu.cn) website. Researchers can request access to the data by registering for a free CHARLS account and submitting a data application form. The CHARLS team will review and approve the application based on the research purpose and feasibility.

**Funding:** The author(s) received no specific funding for this work.

**Competing interests:** The authors have declared that no competing interests exist.

# 1 Introduction

In recent years, depression has become a major global health issue that has received a lot of attention [1]. Depression is one of the most common mental illnesses among middle-aged and older people in China [2]. A large-scale survey on the mental health of the elderly in China found a depression detection rate of 24.0% [3]. The early manifestations of depression frequently present as depressive symptoms. It is, therefore, crucial to recognize these symptoms at an early stage to prevent and control depression in middle-aged and older adults. Depressive symptoms are associated with a variety of unwelcome outcomes including cognitive decline, physical disability, mortality, and medical spending [4,5]. These consequences seriously affect the quality of life and well-being of middle-aged and older adults. The high impact of depression symptoms calls for the identification of early and easily assessable risk factors for its prevention.

Self-rated health (SRH), individuals' subjective assessment of their health status, reflects the individual's active cognitive process [6]. SRH has been a significant predictor of various aspects of current health status, which mirrors biological, social, and environmental factors, as well as cultural and personal beliefs [7]. In several recent studies, researchers have established links between SRH and depressive symptoms and shown that SRH may be an important predictor of depression. A study has shown older adults who rate their health as poor have a higher prevalence of depressive symptoms than those who rate their health as good [8]. Self-rated health remained a predictor of major depressive syndrome for up to 5 years as noted in a study of 789 people with a history of depressive symptoms from a population-based longitudinal health survey [9].

There is also evidence for the effect of depressive symptoms on SRH. A longitudinal inquiry found that higher depressive symptoms predict lower SRH and a decline in SRH at follow-up among Chinese people aged 45 and above [10]. From a physiological point of view, people with depression often experience sleep disturbances, irritability, loss of interest, concentration, and fatigue [11–14]. These symptoms can lead them to believe that their health is even worse, which in turn worsens their depression. From a psychological point of view, individuals affected by depression may feel helpless, hopeless, and anxious, and thus assess their health more pessimistically [15]. Therefore, there may be a reciprocal relationship between depression and SRH.

Urban-rural disparities are critical factors to consider when examining the relationships between SRH and depressive symptoms among middle-aged and older adults. The disparity in depression between urban and rural populations has long been a prominent research focus. Previous studies have consistently demonstrated that the prevalence of depression among older adults in rural China is significantly higher than that of their urban counterparts [16]. Evidence suggests that this elevated risk of depression in rural older adults may be attributed to socioeconomic factors, healthcare systems and policies, levels of social support, and the degree of social participation [17,18]. Similarly, significant urban-rural disparities exist in self-rated health among older adults [19]. Research has shown that older adults in urban

China report slightly better SRH compared to their rural counterparts [20].Understanding whether the bidirectional relationship between SRH and depressive symptoms varies by urban-rural disparities could provide more targeted and actionable strategies for improving the mental health of middle-aged and older adults.

In summary, the mechanisms underlying the relationship between SRH and depressive symptoms are not unidirectional, but complex [21]. To date, few empirical studies have examined the reciprocal relationship between SRH and depressive symptoms. Some studies used a cross-sectional design and were unable to determine a causal relationship between SRH and depression symptoms [22]. Several longitudinal studies that were conducted focused only on one possible direction, either predicting SRH by depressive symptoms or depressive symptoms by SRH [10,23,24]. This research addresses these shortcomings by using a 5-wave longitudinal design analyzed using a cross-lagged panel model, which allows researchers to make stronger correlation inferences and test for effects in both directions simultaneously. We aimed to examine whether SRH in the current wave predicts subsequent depressive symptoms, and whether depressive symptoms in the current wave predict subsequent SRH.

## 2 Materials and methods

### 2.1 Participants and procedures

We used data from the China Health and Retirement Longitudinal Study (CHARLS), which surveyed people aged 45 and above in 150 county-level units and 450 communities in 28 provinces of China. It is designed to analyze population aging and facilitate interdisciplinary research on aging. The baseline survey was conducted in 2011 and follow-up studies were conducted in 2013, 2015, 2018, and 2020. The survey questionnaire was developed based on international sources, and the data quality is widely recognized internationally. To ensure fairness and representativeness in sample selection, CHARLS employs a four-stage sampling process: county (district), village (residential area), household, and individual levels. Specifically, for sampling at the county (district) and village (residential area) levels, CHARLS uses probability sampling proportional to population size [25]. Strict quality control measures are implemented throughout the data collection process, including in phases such as interviewer training and field visits, as well as real-time coding of critical information to ensure standardized visits and data accuracy. All data collected by the project are maintained by the Institute of Social Science Survey at Peking University and are published on the CHARLS project website (http://charls.pku.edu.cn). The database is accessible upon registration and approval. CHARLS has been approved by the Biomedical Ethics Review Committee of Peking University, China (IRB00001052–11015). Finally, all participants provided written informed consent.

In this study, we analyzed longitudinal data on the relationship between SRH and depressive symptoms among middle-aged and older adults in China using five waves of data from 2011 (T1), 2013 (T2), 2015 (T3), 2018 (T4), and 2020 (T5). As shown in Fig 1, a total of 17708 respondents participated at baseline. We excluded 7205 participants due to: 1) under the age of 45 years old (N = 508); 2) individuals who were not successfully followed-up in 2013, 2015, 2018, and 2020 (N = 6596); 3) more than half of the data missing (N = 101). Ultimately, 10503 Chinese middle aged and older adults were included in our study. For the remaining samples, variables with a small number of missing values were handled as follows: (1) For time-invariant variables, if these variables were not missing across all five surveys, data from one of the surveys were used for imputation; (2) For time-variant variables, multiple imputation methods were applied [26].

### 2.2 Measure

**2.2.1 Depressive symptoms.** In CHARLS, depressive symptoms were measured by the Center for Epidemiologic Studies Depression Scale (CESD-10) [27]. This scale provides ten items, including eight negative emotion items and two positive emotion items. The 10 items were scored on a 4-point Likert scale (rarely or not at all = 0, not too much = 1, sometimes or half the time = 2, and most of the time = 3), and the two positive emotion questions were reverse-scored, and then the scores for the 10 questions were summed to construct a composite index of depressed emotion in middle-aged and older adults, which ranged from 0 to 30. Furthermore, the cut-off value of 10 on the CESD-10 has been widely

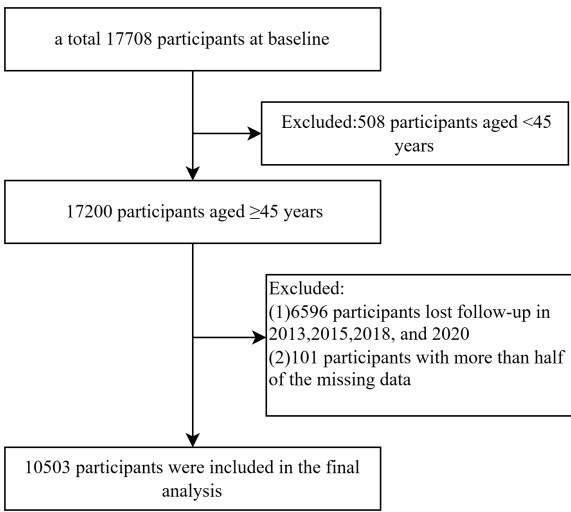

**Fig 1. Flow chart of analytic sample.**

used to define depression in most previous studies [28]. Therefore, in the present study, depressive conditions were dichotomized as non-depressed versus depressed based on the cut-off value of 10. The CESD-10 continuous score and the dichotomous classification of depression can appropriately reflect depression. The CESD-10 continuous score was used for subsequent data analysis in this study.

**2.2.2 Self-rated health.** Self-rated health was evaluated by a single item: "Would you say your health is very good, good, fair, poor,or very poor?" [29]The question was used both at baseline and at the follow-up. The respondents rated their health from 1 (very good) to 5 (very poor). The scale was reverse-coded in the present analysis so that higher ratings indicated better SRH.

**2.2.3 Covariates variables.** Based on existing research, we controlled for the sociodemographic variables, economic status, lifestyle factors, health factors, activity of daily living (ADL), instrumental activity of daily living (IADL), and social support on SRH and depressive symptoms in middle-aged and older adults. Sociodemographic variables included age, gender (0 = male, 1 = female), educational level (0 = primary school and below, 1 = middle school and above), marital status (0 = no spouse, 1 = spouse), residence (1 = rural, 0 = urban),and retirement (1 = yes, 0 = no). Per capita household expenditure as a measure of economic status [30]. Lifestyle factors included smoking and drinking alcohol. Smoking status was defined as current smoker and non-smoker. Drinking status was defined as current drinker and non-drinker. Health factors included whether suffering from chronic diseases (0 = no a chronic disease, 1 = suffering from a chronic disease). ADL asks respondents if they have difficulty with six daily activities such as bathing, toileting, dressing, and eating. If they reported having difficulty completing any item, it was defined as limited ADL. IADL include household chores, preparing hot meals, shopping for groceries, making telephone calls, and taking medications [31]. If they reported having difficulty completing any item, it was defined as limited IADL. Therefore, ADL and IADL were considered to be a binary variable (0 = no limited, 1 = limited).

In CHARLS, the measurement methods for social support currently lack a unified consensus. Drawing on previous research findings, this study categorizes social support into formal and informal types [32,33]. Formal support is provided by the government and includes medical insurance and pension insurance. If respondents participated in any public medical insurance program (including Urban Employee Medical Insurance, Urban and Rural Resident Medical Insurance, Urban Resident Medical Insurance, New Rural Cooperative Medical Insurance, Government Medical Insurance, Medical Assistance, or Urban Non-Employed Medical Insurance), the medical insurance variable was coded as 1. If respondents

are currently receiving, expected to receive in the future, or currently contributing to government pensions, public institution pensions, or basic employee pension insurance, the pension variable is coded as 1.

The two major informal support variables considered are intergenerational support and social participation [34]. Intergenerational support is measured from three aspects: economic support, emotional support, and daily care [35]. Emotional support is measured by the frequency of meetings and contact between middle-aged and older adults and their children. Economic support is assessed based on the amount of financial assistance provided by children to respondents in the past year. Daily care is evaluated by whether respondents live with their children (1 = yes, 0 = no). Social participation is measured by the frequency and occurrence of social activities. Respondents were asked whether they had participated in any of eight types of social activities in the past month [36]. If none were attended, a score of 0 was assigned. For those who participated in any activity, they were further asked about the frequency of participation, categorized as infrequent, almost weekly, or almost daily, with values ranging from 1 to 3. The total score for the eight activities represents the respondent's level of social participation, ranging from 0 to 24, where higher scores indicate greater levels of social participation. In the cross-lagged model, the logarithms of per capita household expenditure and economic support amount are used.

### 2.3 Statistical analyses

First, we conducted a stationary test to ensure the relationship between variables remains constant over time. Second, we performed descriptive statistics for each variable in the baseline period of 2011. Continuous variables were described using mean and standard deviation or median (25th–75th percentile), and count (proportion) was expressed for categorical variables. Third, we presented the relationship between self-rated health and depressive symptoms across different waves by calculating Spearman correlation coefficients. At last, cross-lagged panel models were designed to investigate the longitudinal reciprocal association between SRH and depressive symptoms at five-time points. Paths solved simultaneously within the model include autoregressive paths (the association of one variable with itself at a later point), covariance paths (the correlation between two variables at each point), and cross-lagged paths (the predictive relationship of a variable with another variable at a later point). For covariates, gender, residence, and education level were considered as time-invariant variables, and per capita household expenditure, retirement, pension insurance, health insurance, social participation, intergenerational support, age, marital status, smoking, drinking alcohol, ADL, IADL, and chronic disease were allowed as time-variant covariates. We used the Comparative Fit Index (CFI), Root Mean Square Error of Approximation (RMSEA), and Standardized Root Mean Square Residual (SRMR) for the assessment of model fit. When CFI values >0.90, and RMSEA and SRMR values <0.06, it indicates that the model fit is acceptable [37]. Due to the non-normality distribution of depressive symptoms, the model is estimated using robust maximum likelihood estimation (MLR) [38].

To examine whether there are differences in the longitudinal bidirectional relationship between self-rated health and depressive symptoms among middle-aged and older adults in urban and rural China, a multi-group analysis was conducted. A chi-square difference test was performed to compare the unconstrained and constrained models, with a p-value of less than 0.05 indicating a statistically significant difference [39].

## 3 Results

### 3.1 Stationary test

To avoid spurious regressions and ensure the validity of the estimation results, the study conducts a stationarity test on the panel data. We use the Harris-Tsavalis (HT) test, if the result rejects the original hypothesis of the existence of a unit root, we can consider this series is stationarity, and vice versa, it is not stationarity. The results of the HT test are shown in Table 1, the P-value of both SRH and depressive symptoms is less than 0.001, which strongly rejects the original hypothesis of the existence of a unit root, so we believe that SRH and depressive symptoms is stationarity.

**Table 1. Stationarity test.**

| Variable | Statistic | z | p-value |
|---|---|---|---|
| SRH | −0.482 | −76.333 | 0.000 |
| depressive symptoms | −0.486 | −77.117 | 0.000 |

### 3.2 Basic characteristics of the study population at T1

Table 2 provides data on study participants at T1. The median age of participants was 57 years (IQR: 51–63 years)., of which 53.4% were female and 90.1% of the middle-aged and older adults had a spouse. 66.4% of the middle-aged and older adults lived in rural locations and 67.8% had a primary school or below education.29.6% of them were current smokers and 33.3% were current drinkers. Most of them are not limited to ADL and IADL, accounting for 85.2% and 80.8%, respectively. The median depression symptoms score at baseline was7(IQR: 3–12) and the mean SRH score was 2.99(SD = 0.903).

### 3.3 Spearman correlation between SHR and depressive symptoms

Table 3 presents the Spearman correlation coefficients between SRH and depressive symptoms. At each time point and across time, a negative correlation was observed between SRH and depressive symptoms. In other words, SRH in the current wave had a negative correlation with current depressive symptoms. SRH in the previous wave was negatively correlated with subsequent depressive symptoms, and similarly, depressive symptoms in the previous wave and subsequent SRH were also negatively correlated (all $p < 0.01$). The negative relationship between SRH and depressive symptoms indicate individuals with higher SRH tend to have lower depressive symptoms; conversely, individuals with lower SRH may experience higher depressive symptoms. More detailed information about the correlation coefficient of SRH and depressive symptoms at T1 to T5 can be obtained from Table 3.

### 3.4 The cross-lagged model

We tested the relationship between SRH and depressive symptoms in middle-aged and older adults by constructing a cross-lagged model. At each time point, age, gender, marital status, education level, residence, smoking, drinking alcohol, chronic diseases, per capita household expenditure, retirement, pension insurance, health insurance, social participation, intergenerational support, ADL, and IADL were included as covariates in the model. The results indicate that the model fits well (RMSEA = 0.026, CFI = 0.900, SRMR = 0.036). Table 4 presents the path coefficients for the cross-lagged model. The autoregressive paths of both variables from T1 to T5 were significant, indicating that baseline levels of SRH and depressive symptoms were predictive of the same variables at follow-up. The cross-lagged paths from SRH to depressive symptoms were significant ($\beta_{(T1-T2)} = -0.503, p < 0.001$, $\beta_{(T2-T3)} = -0.564, p < 0.001$, $\beta_{(T3-T4)} = -0.405, p < 0.001$, $\beta_{(T4-T5)} = -0.441, p < 0.001$). The paths from depressive symptoms to SRH were significant($\beta_{(T1-T2)} = -0.015, p < 0.001$, $\beta_{(T2-T3)} = -0.018, p < 0.001$, $\beta_{(T3-T4)} = -0.014, p < 0.001$, $\beta_{(T4-T5)} = -0.014, p < 0.001$).

### 3.5 Multi-group cross-lagged panel model

To further assess the potential impact of urban-rural disparities on the cross-lagged model among middle-aged and older adults, a multi-group analysis procedure was conducted. In this analysis, the data were divided into two groups (e.g., urban and rural). First, an unconstrained model was established, allowing all paths to be freely estimated across the urban and rural groups. Subsequently, a constrained model was constructed in which autoregressive paths and cross-lagged paths were set to be equal across the urban and rural groups. Next, we compared the two models to examine possible differences between them ($\Delta\chi^2 = 11.8$, $\Delta df = 16, p = 0.758$). The result indicates that the association between self-rated

   

**Table 2. Basic characteristics at T1.**

| variable | | | |
|---|---|---|---|
| continuous variable | | | |
| Depressive symptoms, Median (IQR) | 7(3,12) | | |
| SRH, mean (SD) | 2.99(0.903) | | |
| Age (year), Median (IQR) | 57(51,63) | | |
| Economic Status (yuan), Median (IQR) | 5302.86(2987,9770) | | |
| Financial Support (yuan), Median (IQR) | 0(0,800) | | |
| social participation, Median (IQR) | 1(0,3) | | |
| categorical variables | values | N | Percentage (%) |
| Depressive symptoms | non-depressed (0) | 6620 | 63.03 |
| | depressed (1) | 3883 | 36.97 |
| gender | male ( 1 ) | 4899 | 46.6 |
| | female ( 0 ) | 5604 | 53.4 |
| residence | rural (1) | 6970 | 66.4 |
| | urban (0) | 3533 | 33.6 |
| Education level | primary school and below (0) | 7123 | 67.8 |
| | middle school and above (1) | 3380 | 32.2 |
| marital status | spouse ( 1 ) | 9462 | 90.1 |
| | no spouse ( 0 ) | 1041 | 9.9 |
| Retirement | Yes (1) | 983 | 9.4 |
| | No (0) | 9520 | 90.6 |
| Pension insurance | Yes (1) | 3533 | 33.6 |
| | No (0) | 6970 | 66.4 |
| Health insurance | Yes (1) | 9963 | 94.9 |
| | No (0) | 540 | 5.1 |
| Daily care | Yes (1) | 1535 | 14.6 |
| | No (0) | 8968 | 85.4 |
| Emotion support | Once every few months (1) | 460 | 4.4 |
| | Once or twice a month (2) | 2097 | 20 |
| | At least once a week (3) | 7946 | 75.7 |
| smoking | current smoker ( 1 ) | 3104 | 29.6 |
| | non-smoker ( 0 ) | 7399 | 70.4 |
| drinking alcohol | current drinker ( 1 ) | 3500 | 33.3 |
| | non-drinker ( 0 ) | 7003 | 66.7 |
| chronic diseases | yes ( 1 ) | 7041 | 67 |
| | no ( 0 ) | 3462 | 33 |
| ADL | no limitations ( 0 ) | 8945 | 85.2 |
| | at least one limitations (1) | 1558 | 14.8 |
| IADL | no limitations ( 0 ) | 8487 | 80.8 |
| | at least one limitations (1) | 2016 | 19.2 |

Note: N = 10503. M = mean; SD = Standard Deviation; ADL = activity of daily living; IADL = instrumental activity of daily living; IQR = Interquartile Range.

**Table 3. Spearman correlation coefficients between SRH and depression symptoms.**

| | SHR(T1) | DS(T1) | SHR (T2) | DS(T2) | SHR(T3) | DS(T3) | SHR(T4) | DS(T4) | SHR (T5) | DS(T5) |
|---|---|---|---|---|---|---|---|---|---|---|
| SHR(T1) | 1 | | | | | | | | | |
| DS(T1) | −0.415** | 1 | | | | | | | | |
| SHR (T2) | 0.434** | −0.307** | 1 | | | | | | | |
| DS(T2) | −0.303** | 0.440** | −0.389** | 1 | | | | | | |
| SHR(T3) | 0.390** | −0.290** | 0.456** | −0.319** | 1 | | | | | |
| DS(T3) | −0.293** | 0.437** | −0.316** | 0.486** | −0.391** | 1 | | | | |
| SHR(T4) | 0.362** | −0.279** | 0.414** | −0.294** | 0.472** | −0.319** | 1 | | | |
| DS(T4) | −0.265** | 0.383** | −0.281** | 0.435** | −0.294** | 0.477** | −0.377** | 1 | | |
| SHR (T5) | 0.329** | −0.262** | 0.362** | −0.273** | 0.425** | −0.305** | 0.495** | −0.312** | 1 | |
| DS(T5) | −0.260** | 0.383** | −0.273** | 0.410** | −0.280** | 0.456** | −0.315** | 0.485** | −0.402** | 1 |

Note:

**$p < 0.01$. SRH = self-rated health; DS = depressive symptoms.

**Table 4. Cross-lagged model path coefficient.**

| Regression Path | T1-T2 | T2-T3 | T3-T4 | T4-T5 |
|---|---|---|---|---|
| SRH->DS | −0.507***(0.061) | −0.564***(0.061) | −0.405***(0.062) | −0.441***(0.059) |
| DS ->SRH | −0.015***(0.001) | −0.018***(0.002) | −0.014***(0.001) | −0.014***(0.001) |
| SRH->SRH | 0.295***(0.011) | 0.33***(0.011) | 0.341***(0.011) | 0.339***(0.010) |
| DS->DS | 0.312***(0.010) | 0.353***(0.012) | 0.371***(0.010) | 0.315***(0.010) |

Note:

***$p < 0.001$. SRH = self-rated health; DS = depressive symptoms.

health and depressive symptoms does not differ between urban and rural, suggesting that urban-rural disparities does not moderate the relationship between SRH and depressive symptoms.

## 4 Discussion

The purpose of this study was to clarify the longitudinal and bidirectional relationship between SRH and depressive symptoms among middle-aged and older adults. Therefore, based on data from CHARLS, we employed a cross-lagged panel analysis that examined these associations longitudinally. A cross-lagged design made it possible to study SRH on subsequent depression symptoms and depression symptoms on subsequent SRH while controlling the previous level of each variable. The findings indicated that SRH at Time 1 significantly and negatively predicted depression symptoms at Time 2, Time 2 SRH leading to Time 3 depression symptoms, as well as Time 3 SRH leading to Time 4 depression symptoms and Time 4 SRH leading to Time 5 depression symptoms, and vice versa. These suggest that, among middle-aged and older adults, worse SRH leads to higher levels of depressive symptoms and higher levels of depressive symptoms lead to worse SRH. In brief, the change in depression symptoms could be appropriately reflected by the change in SRH in middle-aged and older adults. Improving SRH may be worthwhile for monitoring and managing depressive symptoms in the middle-aged and elderly population.

The results of this study are consistent with those of previous studies examining the longitudinal relationship between SRH and depression symptoms [22,40,41]. In the Survey of Health, Ageing and Retirement in Europe of 28899 participants, worse Self-rated health leads to higher levels of depressive symptoms and higher levels of depressive symptoms lead to worse self-rated health among middle-aged and older adults [40]. A cross-sectional study of 5119 participants

from the China Longitudinal Healthy Longevity Survey identifies that self-rated health has direct or indirect associations with depressive symptoms in older people [22]. In addition, the results of our analysis indicate that the path from SRH to depressive symptoms is more robust than the path from depressive symptoms to SRH. Perhaps because SRH is influenced by physical health, psychological, lifestyle, and sociodemographic factors [42]. These factors may contribute to the development of depressive symptoms, and the presence of multiple factors may result in a greater effect. Studies have shown that objective health threats such as coronary heart disease and chronic disease increase the risk of depression [43,44]. This suggests that SRH may be able to predict the onset of depressive symptoms by reflecting changes in objective health indicators.

This study supports the negative relationship between SRH and depressive symptoms in middle-aged older adults. As other researchers have previously discussed, the better a person's self-rated health, the more likely they are to adopt a positive attitude towards life such as expressing emotions calmly or thinking positively, and the less likely they are to become depressed [45]. A longitudinal study of participants aged 40 and above from the German Ageing Survey suggests that worsening self-rated health is associated with a decrease in positive affect and an increase in negative affect, which are typical hallmarks of depressive symptoms [46,47]. It also reveals the negative relationship of depressive symptoms on SRH. The research has indicated a correlation between depression and an increased risk of developing chronic diseases and experiencing cognitive decline [48]. Consequently, middle-aged and older adults with depression are more prone to adverse health outcomes, which may further impede their ability to perceive their health status positively.

We found no urban-rural difference in the bidirectional associations between SRH and depressive symptoms. This means that the links between SRH and depressive symptoms may be similar in both urban and rural China. We identified two possible reasons for this result. First, the age structure, gender distribution, and prevalence of chronic diseases among middle-aged and older adults in urban and rural areas are relatively similar. These factors may weaken the differences in the relationship between health and depression across urban and rural settings. Second, there may be other potential factors, such as psychological factors, that influence the relationship between self-rated health and depression among middle-aged and older adults in both urban and rural areas [49].

Several limitations should be acknowledged in the present study. First, the presence of depressive symptoms was determined through self-reporting by the participants, and it should be noted that a medical diagnosis of depression was not obtained. There may be a degree of reporting bias [50]. Second, the effect size of the regression path from depressive symptoms to self-rated health is small, and further analysis is required to ascertain whether there are factors that influence the self-rated health and depressive symptoms of middle-aged and elderly individuals. Even though this study controlled for age, gender, marital status, education level, residence, smoking, drinking alcohol, chronic diseases, per capita household expenditure, retirement, pension insurance, health insurance, social participation, intergenerational support, ADL, and IADL to minimize the impact of confounding factors, it is challenging to control for all confounding factors through control variables. Third, the discussion of the mechanism of the bidirectional relationship between SRH and depressive symptoms is inadequate. In subsequent studies, we will continue to discuss factors for their possible mediating effects and explore whether factors such as gender and age groups play a moderating role in the bidirectional relationship between SRH and depressive symptoms.

## 5 Conclusion

In Chinese middle-aged and older adults, SRH is associated with depressive symptoms. Specifically, SRH in the current wave had a negative correlation with current depressive symptoms. SRH in the previous wave was negatively correlated with subsequent depressive symptoms. Depressive symptoms in the previous wave and subsequent SRH were also negatively correlated. Therefore, either SRH improvement or depression prevention for middle-aged and older adults should be targeted at each other. It may be worthwhile to Interventions to improve depression are combined with strategies to improve self-rated health. For middle-aged and older adults, it is valuable to combine interventions to improve depression

with strategies to improve self-rated health. Additionally, we found the cross-lagged effects between self-rated health and depressive symptoms did not differ by urban-rural. This suggests that health and psychological interventions for middle-aged and older adults can be designed with a high degree of generalizability.

## Acknowledgments

Thanks to China Social Science Research Center of Peking University and its personnel for data assistance.

## Author contributions

**Conceptualization:** Shuyan Leng, Lihua Yao.

**Data curation:** Shuyan Leng, Jiajia Deng.

**Formal analysis:** Shuyan Leng.

**Funding acquisition:** Lihua Yao.

**Investigation:** Lihua Yao.

**Methodology:** Shuyan Leng.

**Resources:** Lihua Yao.

**Supervision:** Lihua Yao.

**Validation:** Shuyan Leng.

**Visualization:** Shuyan Leng.

**Writing – original draft:** Shuyan Leng.

**Writing – review & editing:** Jiajia Deng.

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
