## [Decision Letter · Decision Letter 0]

25 Dec 2024

PONE-D-24-54916Associations between self-rated health and depressive symptoms among middle-aged and older adults: A cross-lagged panel analysis2011-2020PLOS ONE

Dear Dr. Yao,

Thank you for submitting your manuscript to PLOS ONE. After careful consideration, we feel that it has merit but does not fully meet PLOS ONE’s publication criteria as it currently stands. Therefore, we invite you to submit a revised version of the manuscript that addresses the points raised during the review process.

We look forward to receiving your revised manuscript.

Kind regards,

Zhuo Chen, Ph.D.

Academic Editor

PLOS ONE

Journal Requirements:

2. Please note that your Data Availability Statement is currently missing a direct link to access each database. If your manuscript is accepted for publication, you will be asked to provide these details on a very short timeline. We therefore suggest that you provide this information now, though we will not hold up the peer review process if you are unable.

Reviewers' comments:

Reviewer's Responses to Questions

**Comments to the Author**

1. Is the manuscript technically sound, and do the data support the conclusions?

Reviewer #1: Yes

Reviewer #2: Yes

2. Has the statistical analysis been performed appropriately and rigorously? 

Reviewer #1: No

Reviewer #2: No

3. Have the authors made all data underlying the findings in their manuscript fully available?

Reviewer #1: Yes

Reviewer #2: Yes

4. Is the manuscript presented in an intelligible fashion and written in standard English?

Reviewer #1: Yes

Reviewer #2: Yes

5. Review Comments to the Author

Reviewer #1: The research topic is interesting, examining associations between self-rated health (SRH) and depressive symptoms among middle-aged and older adults using a cross-lagged panel analysis. However, I have a few suggestions and concerns:

1. It may be beneficial to specify the geographical context in the title. For example, adding "in China" would provide clarity and relevance.

2. References (Lines 61–67): The statements made in these lines should be supported by appropriate references. Please cite relevant literature to substantiate your claims.

3. Sample Selection: I am slightly concerned about the sample selection process. It appears that the analysis only included participants who were present in all five waves of data collection. Could the authors clarify whether this approach might introduce bias and how it affects the representativeness of the sample?

4. Methodological Considerations:

Social Support: As social support is a well-known factor influencing both depressive symptoms and self-rated health, it would be prudent to include it.

Economic Status: Economic conditions are another important determinant of both mental and physical health. Please consider including economic status in the analysis or discussing its potential influence on the findings.

Correlation Coefficients: The manuscript should specify the types of correlation coefficients used (e.g., Pearson, Spearman) to ensure clarity and reproducibility.

5. Table (M, SD, Mis): When reporting descriptive statistics such as M and SD, please clarify whether the data are presented as mean and standard deviation, or if a median and interquartile range might be more appropriate given the data distribution.

6. Interpretation of Findings (Lines Discussing "Effect"): The manuscript states that the study supports a "negative predictive effect" of SRH on depressive symptoms. However, as this is not an interventional study, it may be more accurate to use terms such as "association" or "relationship" rather than "effect," which implies causality.

Reviewer #2: This paper fills the gap in the existing literature by exploring the reciprocal relationship between depressive symptoms and self-rated health (SRH) among Chinese middle-aged and older adults by calculating correlation coefficients and using cross-lagged panel model. The paper highlights two main findings. First, self-rate health can be used to predict depressive symptoms in the subsequent period and vice versa. Second, the pathway from SRH to depressive symptoms is more robust than the reverse pathway.

Below are my major comments:

1. The authors may be more careful when using causal inference. In line 69 and line 70, the authors state that the longitudinal design strengthens causal inference. However, there are three limitations to establish causality. Firstly, there might be unobserved factors such as personal traits and caregiving decision that may simultaneously influence both SRH and depressive symptoms. Secondly, both SRH and depressive symptoms are self-reported, potentially leading to inaccuracies. Finally, the authors do not test for stationarity (I mention it in my second point below) of variables. If the stationary assumption is violated, the causal relationship may no longer hold. Thus, it would be better to replace stronger causal inference with stronger correlation.

2. The authors may find it useful to conduct a stationary test to ensure the relationship between variables remains constant over time. If the data is non-stationary, appropriate transformations or differencing may be required to avoid spurious regression between SRH and depressive symptoms. For instance, changes in SRH or depressive symptoms resulting from structural shifts, such as population aging or the COVID-19 pandemic, cannot conclusively establish a causal relationship.

3. The authors might consider adding more time-variant covariates such as employment status or income. The dataset includes individuals aged 45 and older, many of whom may have retired during the survey period. Employment status (or income) could affect SRH and depressive symptoms for two reasons. On one hand, retirement may lead to feeling of upset as individuals may find it difficult to recognize their value as they are no longer needed due to transition out of professional roles. On the other hand, Reduced income after retirement may result in financial pressures, particularly for individuals who still have to support their children or grandchildren.

4. It is better if the authors could be clearer about the variables used and reported. In Line 113-115, the authors categorized the CESD-10 score into a binary variable indicating whether have depressive symptoms or not, but fail to provide descriptive statistics of the binary variable indicating the depressive symptoms. It would be better and clearer to provide descriptive statistics on both depressive symptoms and CESD-10 score. Additionally, the authors should specify which variable (CESD-10 scores or depressive symptoms) is used in the statistical analyses presented in Sections 3.2 and 3.3.

5. In the Section 3.2, the authors could be clearer about the expression. For Table 2 in the Section 3.2, it would be better to explain more about the numbers in the first rows. If my understanding is correct, the numbers in the first row aligns with those in the first column, and it would be better to make it clearer in the notes below.

In addition, in the Section 3.2, the authors could provide more interpretations about the correlation results so that it would be easier to understand. For instance, the negative relationship between SRH and depressive symptoms indicate individuals with higher

6. The authors could consider exploring potential heterogeneous effect. For instance, the path coefficients from SRH to depressive symptoms might vary across age groups, gender, or geographic regions. It may help to enrich the study.

I hope these comments would be helpful.

6. PLOS authors have the option to publish the peer review history of their article (what does this mean? ). If published, this will include your full peer review and any attached files.

**Do you want your identity to be public for this peer review?** For information about this choice, including consent withdrawal, please see our Privacy Policy .

Reviewer #1: No

Reviewer #2: No

---

## [Author Response · Author response to Decision Letter 0]

17 Jan 2025

The reviewer’s comments have been addressed and responded to point by point in the attached document (filename: Response to Reviewers).

---

## [Decision Letter · Decision Letter 1]

5 Mar 2025

Associations between self-rated health and depressive symptoms among middle-aged and older adults in ChinaA cross-lagged panel analysis2011-2020

PONE-D-24-54916R1

Dear Dr. Yao,

We’re pleased to inform you that your manuscript has been judged scientifically suitable for publication and will be formally accepted for publication once it meets all outstanding technical requirements.

Kind regards,

Zhuo Chen, Ph.D.

Academic Editor

PLOS ONE

Additional Editor Comments (optional):

There are some minor copyediting issues -- e.g., inconsistent use of spaces. Authors should work with the proofing team to resolve those issues.

Reviewers' comments:

Reviewer's Responses to Questions

**Comments to the Author**

1. If the authors have adequately addressed your comments raised in a previous round of review and you feel that this manuscript is now acceptable for publication, you may indicate that here to bypass the “Comments to the Author” section, enter your conflict of interest statement in the “Confidential to Editor” section, and submit your "Accept" recommendation.

Reviewer #2: All comments have been addressed

2. Is the manuscript technically sound, and do the data support the conclusions?

Reviewer #2: Yes

3. Has the statistical analysis been performed appropriately and rigorously? 

Reviewer #2: Yes

4. Have the authors made all data underlying the findings in their manuscript fully available?

Reviewer #2: Yes

5. Is the manuscript presented in an intelligible fashion and written in standard English?

Reviewer #2: Yes

6. Review Comments to the Author

Reviewer #2: (No Response)

7. PLOS authors have the option to publish the peer review history of their article (what does this mean? ). If published, this will include your full peer review and any attached files.

**Do you want your identity to be public for this peer review?** For information about this choice, including consent withdrawal, please see our Privacy Policy .

Reviewer #2: No

---

## [Editor Report · Acceptance letter]

PONE-D-24-54916R1

PLOS ONE

Dear Dr. Yao,

I'm pleased to inform you that your manuscript has been deemed suitable for publication in PLOS ONE. Congratulations! Your manuscript is now being handed over to our production team.

Kind regards,

on behalf of

Prof. Zhuo Chen

Academic Editor

PLOS ONE